# Influence of the COVID-19 pandemic on running behaviors, motives, and running-related injury: A one-year follow-up survey

**Alexandra F. DeJong Lempke**[1,2,3]\*, **Jay Hertel**[3]

**1** Division of Sports Medicine, Boston Children's Hospital, Boston, Massachusetts, United States of America,
**2** Micheli Center for Sports Injury Prevention, Waltham, Massachusetts, United States of America,
**3** Exercise and Sport Injury Laboratory, Charlottesville, Virginia, United States of America

\* Alexandra.dejong@childrens.harvard.edu

## Abstract

The purpose of this study was to compare running behaviors, motives, and injury during the first full year of the pandemic compared to the year prior, and month of eased restrictions. 466 runners responded to this follow-up survey. Paired t-tests were used to compare time-points. Logistic regressions were used to assess demographic influences on behaviors. During the first full year of the pandemic, runners were more likely to increase their weekly runs (Mean Differences [MD]: 0.29±0.10, p < .001), yet had fewer motives (MD: -0.20±0.06, p = .02) compared to the month of eased restrictions. 18–25-year-olds were most likely to increase running volume (Odds Ratio [OR]: 2.79 [1.06, 7.32], p = .04) during the first full year of the pandemic compared to the year prior. Inexperienced runners reported fewer motives (OR: 3.00–4.41, p = .02-.04), and reduced sustained runs (OR: 2.36 [1.13, 4.96], p = .02) during the first full year of the pandemic compared to the year prior. Inexperienced runners and runners who lost access to preferred locations were at increased injury risk (OR: 1.98–2.63, p: .01-.04) during the first full year of the pandemic compared to the year prior. Our findings highlight pandemic-related changes among the running community that are likely to influence behaviors and injury risk.

## Introduction

The severe acute respiratory syndrome coronavirus 2 (COVID-19) pandemic had a profound impact on daily functioning worldwide due to social isolation and distancing mandates, facility closures, and other locally-imposed guidelines to reduce the spread of the virus. These restrictions led to extensive alterations in physical activity, especially for group sports and other activities that required specific equipment and access to training facilities. However, other forms of individualized physical activity that could be conducted outdoors and with minimal necessary resources, namely running, offered an opportunity for individuals to remain physically active during the pandemic. As such, it is unsurprising that running remains one of the most popular forms of physical activity globally, attracting individuals of all ability levels to engage in this form of moderate-to-rigorous cardiovascular exercise [1, 2].

**Data Availability Statement:** Per the University of Virginia Institutional Review Board, data cannot be made public due to the fact that the consent form indicated that data would not be shared. As such,

we cannot make data publicly available. Requests for data can be sent to Jeffrey Monroe (mjm6ny@virginia.edu), the point of contact for the University of Virginia Institutional Review Board.

**Funding:** The authors received no specific funding for this work.

**Competing interests:** The authors have declared that no competing interests exist.

Running for exercise was an encouraged form of physical activity during the pandemic to improve immune defense [3] and protect against COVID-19-related respiratory health complications [4]. In addition to broad physical health benefits, regular moderate-intensity running participation has previously been associated with improved quality of life and mental health [5, 6]. However, the lifestyle changes incurred in the first several months of the pandemic created a shift in running participation and wellness measures among the running community [7–9]. Runners were found to increase running volume with a decreased emphasis on training intensity during the pandemic compared to prior training tendencies [7, 8]. While physical activity during the pandemic was found to combat extreme declines in mental health [10], runners still reported declines in motivation [7], and poorer psychosocial measures that related to increased risk of injury [8].

While the assessment of runners during the first several months of the pandemic were helpful for understanding the initial impact of social isolation measures on behaviors, motives, and running-related injury, restrictions in the United States were extended for a full year given perpetually high case rates (peak new cases: 312,325 on January 8, 2021 vs. 24,366 peak new cases on May 15th during the initial collection; 12.8 times higher than initial peak survey date cases) and limited vaccination availability [11]. It remains unclear how the full social isolation timeframe affected the running community, especially as there were evolving changes with facility and trail availability, race schedules (with adopted virtual platforms), and other key training accessibility factors. The full year timeframe provides an adequate comparison for injury rates during the entire isolation period compared to previous trends [7]. Several earlier surveys reported slightly different running-related injury rates during the pandemic, which may be attributed to the different timeframes utilized to capture injury rates in initial assessments [7–9].

Assessing the lasting impact of the first full year of the pandemic on the running community is warranted to improve our understanding of behavior and motives changes, and the interplay of running behavioral changes on injury burden. Additionally, contextualizing the full year of the pandemic to behaviors captured in the month of eased restrictions would help provide insight into how runners may be re-adjusting their training. The purpose of this study was to conduct a follow-up assessment of running training, behaviors, motives, and running-related injury risk during the full year of the pandemic compared to outcomes documented in the full year prior to the pandemic [7]. Additionally, we compared running behaviors and motives during the month of eased restrictions (May 2021) to the first full year of the pandemic (May 2020-April 2021) to assess any recent shifts in the running community. Finally, we summarized virtual and in-person racing participation trends during the pandemic. We anticipated that runners would report increased running volume, with concomitantly reduced running intensity and fewer training motives compared to the year prior to the pandemic [7]. We hypothesized that running-related injury risk would be higher during the first full year of the pandemic attributed to increased running volume creating more time under tension during training, and overall abrupt changes in running routine behavior. Finally, we anticipated that running volume would decrease yet running intensity would increase in the month of eased restrictions compared to the first full year of the pandemic.

## Materials and methods

This survey was conducted as a follow-up to an initial assessment of running behaviors, motives, and injury trends in the first three months (March-May 2020) of social isolation measures in the United States (May 4th-June 4th, 2020 active survey collection) [7]. Original inclusion criteria required participants to be at least 18 years of age, and currently or previously

participating in running at any experience level within one year of the study. Participants who previously provided written informed consent to study participation and provided a valid email address (N = 1,025) were contacted twice in a one-month period to complete this subsequent assessment the first full year following the initial survey launch date (May 4th- June 4th active follow-up survey collection). Participants that completed the survey upon the initial email were told to disregard the follow-up email encouraging more responses to eliminate duplicate responses. The original study procedures and follow-up survey launch were approved by the University of Virginia's Institutional Review Board for Social and Behavioral Sciences (#3677), and all participants provided written informed consent.

The survey was created in English by two researchers, and piloted among a group of 10 runners to determine face validity and improve question clarity. The follow-up survey asked participants to provide the same email address from which they received the study invitation to ensure responses could be matched to their original response. This follow-up survey mirrored the original survey [7], and consisted of two primary sections asking questions pertaining to the first full year of the pandemic (May 2020 –April 2021) and to the previous month (May 2021) of eased restrictions in the United States. The survey questions focused on running volume (runs per week, sustained runs per week [i.e., steady state running], workouts per week [(i.e., speed intervals, fartleks, tempo runs, hill repetitions, etc.], weekly mileage, number of cross-training activities), behaviors (pace, access to preferred running routes, if they ran with others or on their own, and typical training time block(s) per day, race participation), motives (reasons for running, level of concern on their training), and running-related injuries (number, time-loss, time of modified training, injury types). The complete survey can be found in the S1 File.

## Data processing

Partial or incomplete responses were excluded from statistical analyses, and all complete responses from the follow-up survey were maintained to compare the full year of the pandemic (May 2020 –April 2021) to the month (May 2021) of eased social isolation restrictions.

Email address responses were compared to the original survey sample to determine which participants completed both timepoints, and this study subset was maintained to directly compare responses from the year prior to the pandemic to the first full year of the pandemic. Running behavior data for the responses recorded for both timepoints were binned and categorized as increased (1), decreased (-1), or no change (0) in running volume if there was more than a one-unit change between timepoints to facilitate logistic regression analyses. Mileage was similarly binned and compared within 10 miles per week, and pace within 30 seconds. Participants' demographic data was carried over from the original survey to account for mediating factors influencing running behaviors, and binned for statistical analyses as follows: age (18–25, 26–35, 36–45, 46–55, 56+ years), and years of running experience (0–3, 4–10, 11–15, 16–20).

## Statistical analyses

Summary statistics were used to assess running behaviors and motives for both study timepoints (month of eased restrictions and the first full year of the pandemic), and to assess virtual and in-person race participation. Analyses were separated as not all responses from the current survey could be linked to original survey responses due to differing or missing email addresses. Injury characteristics during the year of the pandemic were assessed to determine the number, type by body part, and duration of modified and time-loss from training during the pandemic.

Preliminary histogram assessments reflected that the data were normally distributed, supporting subsequent parametric statistical analyses. Paired samples t-tests were conducted to

compare running volume, behaviors and number of reported motives and running time blocks per day between the month of eased restrictions and the year of the pandemic, with alpha set *a priori* to .05. The same analyses were conducted comparing the year of the pandemic and the year prior to the pandemic for the subset of respondents who could be matched to original responses. Mean differences and Cohen's d effect sizes were used to assess the magnitude of differences between timepoint comparisons.

Multivariate logistic regression analyses were used to assess the influence of respondent demographics on running behaviors between the year of the pandemic compared to the year prior to the pandemic. Specific outcomes included total number of runs per week, sustained runs, workouts, cross-training, training times of day, and running motives. Demographic factors included in the regression models were biological sex at birth, and binned age and years of running experience variables. While interactions between demographic variables were considered, preliminary analyses suggested there were no statistically significant relationships and thus were not included for the final regression models.

Injury risk ratio (IRR; Equation) was used to compare injuries incurred during the year of the pandemic compared to the year prior to the pandemic. Resultant values greater than 1 were interpreted as increased risk of injury during the pandemic, and values less than 1 as increased injury risk in the year prior to the pandemic.

$$Injury\ Risk\ Ratio = \frac{([Injuries\ Prior\ to\ the\ Pandemic/Total\ Respondents] * 100)}{([Injuries\ During\ the\ Pandemic/Total\ Respondents] * 100)}$$

A binary logistic regression model was used to assess the influence of demographics and training factors, including mileage and number of runs, on injury occurrence during the year of the pandemic, covarying for injuries incurred in the year prior to the pandemic.

## Results

There was a 45.5% response rate from the 1,025 participants contacted for this follow-up survey (N = 466). Of the participants that responded, 413 provided matching e-mail addresses to the original survey to be included for the full year comparisons (sex: 275 F, 138 M; age: 37±13 years [range: 18–75]; experience: 11±6 years [range: 0–20+]).

### Month of eased restrictions compared to the first full year of the pandemic

Summary statistics of participants' running behaviors and motives are presented in Table 1 and Fig 1a–1d. Notably, there were significant differences between timepoints for number of runs, number of workouts, and weekly mileage. Compared to the month of eased restrictions, respondents reported higher number of total runs (Mean Difference [MD]: 0.29±0.10, t = 4.52, $p < .001$; d = 0.12; Table 1), running workouts (MD: 0.19±0.05, t = 4.41, $p < .001$; d = 0.17; Table 1), and higher weekly mileage (MD: 3.88±0.82, t = 7.84, $p < .001$; d = 0.22; Table 1) during the first full year of the pandemic. There were additionally significant differences in the number of reported running time blocks and motives (Table 1), however were comparable across distributions of preferred time blocks and motives (Fig 1a and 1b).

While 40–47% of respondents did not lose access to preferred roads/paths and trails, most participants did lose access to preferred running gyms (99.4%) and tracks (76.4%), and between 28–41% had still not regained access to these running areas at the time of survey completion (Fig 1c). During the first full year of the pandemic, participants reported that their training increased a little through to a great deal (42.06%) compared to the past month of eased restrictions (35.63%, Fig 1d). Of all respondents, approximately 50% reported re-gaining access to preferred gyms, roads/paths, tracks, and trails in the past month of eased restrictions.

**Table 1. Summary statistics of running behaviors during the first full year of the pandemic, and the month of eased restrictions.**

|  | First Full Year of the Pandemic (Mean ± SD) | Month of Eased Restrictions (Mean ± SD) | MD ± SD (ES) | p-Value |
|---|---|---|---|---|
| **Total Runs Per Week (N)** | 4.19±2.14 | 3.90±2.34 | 0.29±0.10 (0.12) | < .001* |
| **Sustained Runs Per Week (N)** | 3.23±1.78 | 3.20±2.00 | 0.03±0.01 (0.01) | 0.20 |
| **Running Workouts Per Week (N)** | 0.91±1.22 | 0.72±1.14 | 0.19±0.05 (0.17) | < .001* |
| **Cross-Training Activities Per Week (N)** | 2.99±2.36 | 3.02±2.64 | 0.03±0.02 (0.01) | 0.71 |
| **Weekly Mileage (mi)** | 24.07±18.29 | 20.19±17.51 | 3.88±0.82 (0.22) | < .001* |
| **Sustained Running Pace (min/mi)** | 9:20±1:33 | 9:21±1:36 | 0:01±0:01 (0.04) | .38 |
| **Workout Running Pace (min/mi)** | 7:44±1:51 | 7:43±1:49 | 0:01±0:01 (0.04) | .62 |
| **Running Motives (N)** | 1.96±1.27 | 2.16±1.27 | -0.20±0.06 (0.16) | .02* |
| **Typical Running Time Blocks Per Day (N)** | 1.83±1.02 | 1.65±0.92 | 0.18±0.04 (0.20) | .01* |

Abbreviations: SD, standard deviation; MD, mean difference; ES, effect size; N, number; mi, miles; min, minutes.

*Significant at p < .05.

Although only 19–21% of respondents were somewhat or very concerned about the impact of the pandemic on their training throughout the past year and month of eased restrictions, about 26–37% reported concerns of the impact of the pandemic on their running goals (Fig 2).

Only 53% of the sample reported being able to return to their preferred runner groups that they had previously been involved in prior to the pandemic (Fig 1c). Approximately 57% of the respondent sample competed across 737 virtual races (N = 266), while only 33% competed across 449 in-person races during the first full year of the pandemic (N = 154). Specific event participation trends can be found in Fig 3.

## Full year comparisons

Respondents were more likely to change their running behaviors during the first full year of the pandemic than to maintain running habits, especially with increased running volume and fewer motives, which were mediated by age and running experience (Table 2).

Runners between the ages of 46 and 55 were more likely to decrease their total number of runs during the year of the pandemic compared to younger runners (Odds Ratios with 95% Confidence Interval $[OR]_{46-55vs.18-25}$: 0.35 [0.12,1.01], p = .05; Table 2), who were more likely to increase their total number of runs than runners 36–45 ($OR_{18-25vs.36-55}$: 2.79 [1.06, 7.32], p = .04; Table 2). Overall, runners were more likely to increase their sustained runs during the pandemic, which was mediated by runners' experience. Runners with 11–15 years of experience were more likely to increase sustained runs compared to more experienced runners during the pandemic ($OR_{0-3vs.11-15}$: 0.42 [0.19,0.94], p = .03; Table 2), while inexperienced runners were more likely to decrease their number of sustained runs ($OR_{0-3vs.4-10}$: 3.00 [0.99, 9.04], p = .05; $OR_{0-3vs.11-15}$: 4.41 [1.32,14.82], p = .02; Table 2).

Runners ages 18–25 were less likely to decrease cross-training activities compared to runners ages 36–45 during the year of the pandemic ($OR_{18-25vs36-45}$: 0.47 [0.22, 1.04], p = .05; Table 2), however overall runners were found to decrease cross-training activities during the pandemic compared to prior behaviors. Younger runners were additionally found to increase the number of time blocks they would run per day compared to older runner groups ($OR_{18-25vs.26-35}$: 2.47 [1.07, 5.67], p = .03; $OR_{18-25vs.36-45}$: 4.15 [1.48, 11.63], p = .01; Table 2), and conversely older runner groups were more likely to decrease the number of running blocks per day ($OR_{18-25vs.26-35}$: 0.30 [0.09, 1.01], p = .05; $OR_{18-25vs.36-45}$: 0.23 [0.05, 0.94], p = .04;

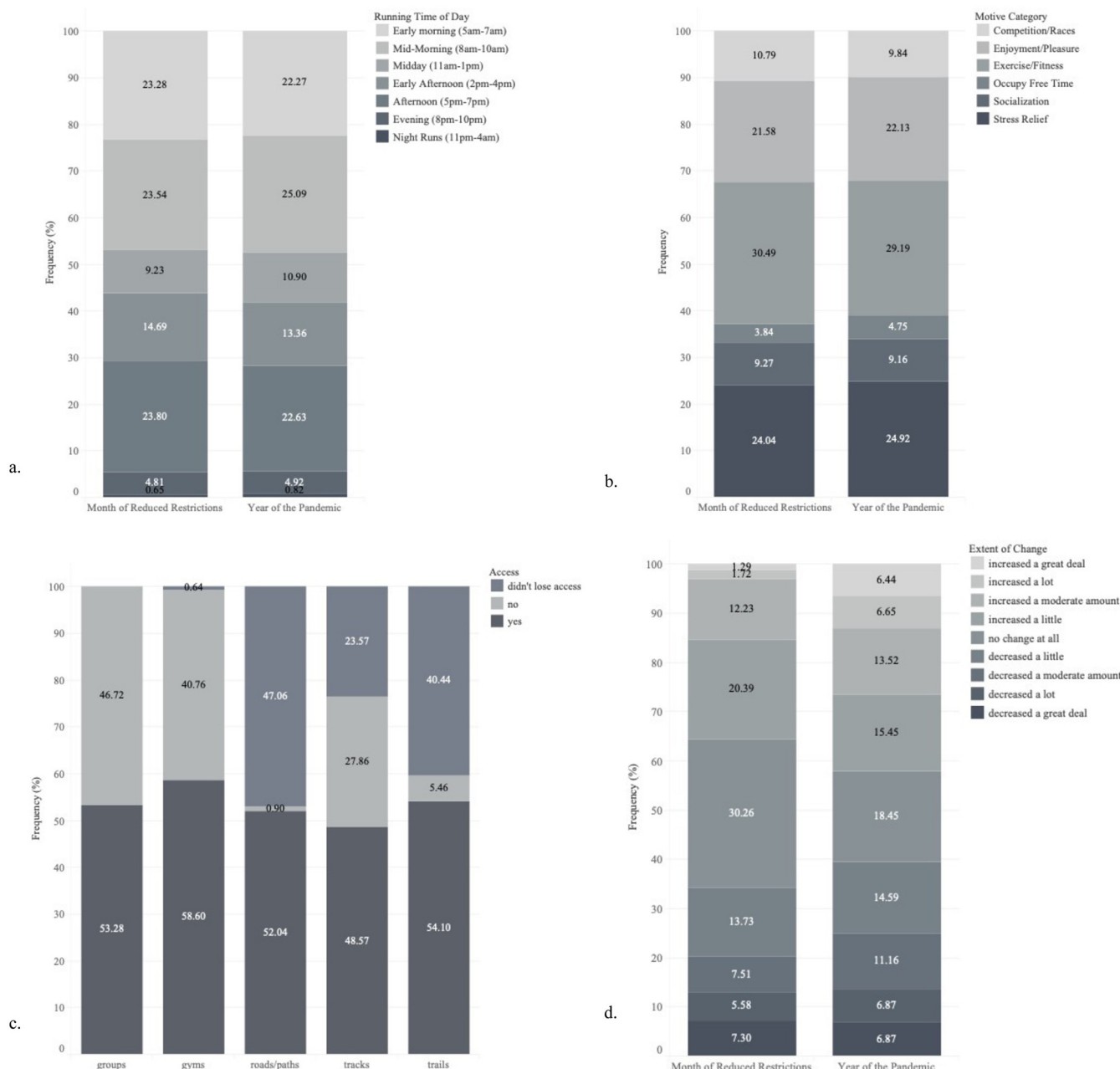

**Fig 1.** Summary of participants' a) running times blocks per day, b) motives for running, c) access to preferred running locations, and d) extent of change in running training during the year of the pandemic compared to the month of eased restrictions. Stacked bar plots comparing the month of eased restrictions (left graphs) to the first full year of the pandemic (right graphs) on participants' subjective responses to their a) preferred times of day that they train, b) motives for running, c) access to preferred running location sites, d) and perceived change in running training. Response percentages are listed by response category overlaid on the bar plots.

Table 2). While runners were more likely to decrease their total number of workouts and increase their weekly mileage during the year of the pandemic, there were no statistically significant demographic factors influencing these changes. Finally, both younger and less experienced runners were most likely to decrease their number of running motives during the year

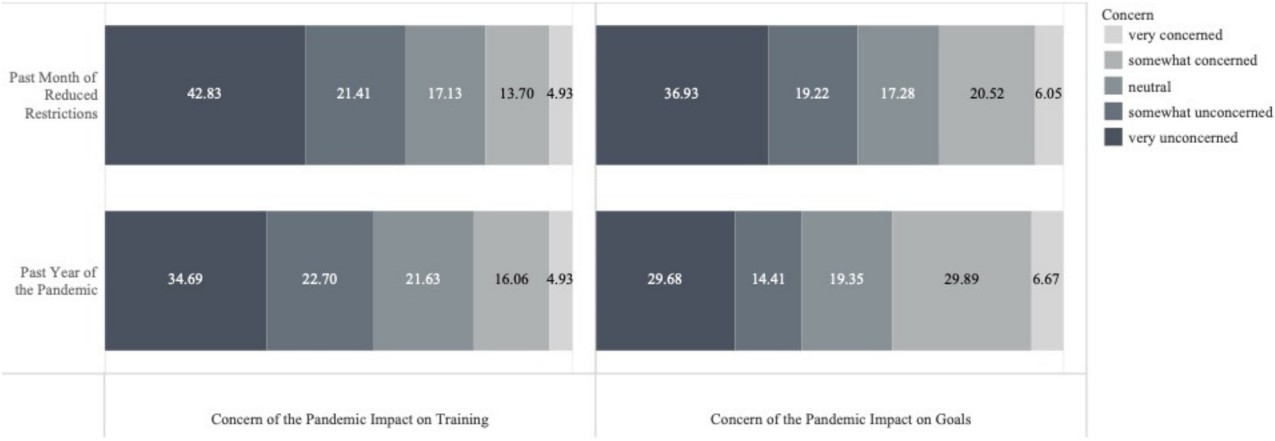

**Fig 2. Level of concern about running training and goals during the pandemic and the past month of reduced restrictions.** Stacked bar plots depicting participants' subjective responses to their level of concern of the pandemic's impact on their running training (left graphs), and their level of concern of the pandemic's impact on their running goals (right graphs), both for the past year of the pandemic (May 2020 –April 2021 [bottom graphs]), and the past month of reduced restrictions (May 2021 [top graphs]). Response percentages are listed by response category overlaid on the bar plots.

of the pandemic ($OR_{18-25vs.36-45}$: 2.02 [1.07, 4.06], p = .05; $OR_{18-25vs.56+}$: 4.69 [1.77, 12.43], p = .002, $OR_{0-3vs16-20}$: 2.36 [1.13, 4.96], p = .02; Table 2).

## Running-related injuries

Approximately one-third of respondents reported sustaining a running-related injury during the first full year of the pandemic (N = 164, 1.6±0.7 injuries per injured runner). The breakdown of injury types by body part can be found in Table 3. When directly comparing the subset of respondents who could be traced to their original responses (N = 413), there was not a significant increase in injury risk during the first full year of the pandemic compared to the year prior to the pandemic (IRR: 1.10 [0.60,1.60]). The average amount of reported time-loss due to injury was higher during the first full year of the pandemic (32±27 days, most often reported 14 days) compared to the year prior (29±24 days, most often reported 5 days; d = 0.13, p = .01). The average time of modified training due to injury was similar across timepoints. When accounting for demographic factors, the odds of sustaining an injury during the first full year of the pandemic was 5.38 times higher than the year prior (4.24 [1.59, 11.30], p = .001). Novice runners ($OR_{0-3vs11-15}$: 2.63 [1.07, 6.46], p = .04), and runners who reported losing access to preferred training environments (OR: 1.98 [1.25, 3.13], p = .01) were more likely to sustain running-related injuries during the first full year of the pandemic.

## Discussion

We sought to identify the influence of the first full year of social isolation restrictions on running training, behaviors, and motives compared to recently eased restrictions, and compared to the year prior to the pandemic. This is the first study to our knowledge that has been able to capture a more complete timeline of the pandemic's influence on physical activity changes in the United States. The training changes identified through this assessment will provide key context for running coaches working with athletes returning to sport following the pandemic, and the motivations and injury risk findings provide clinical insights for patient education and treatment trends.

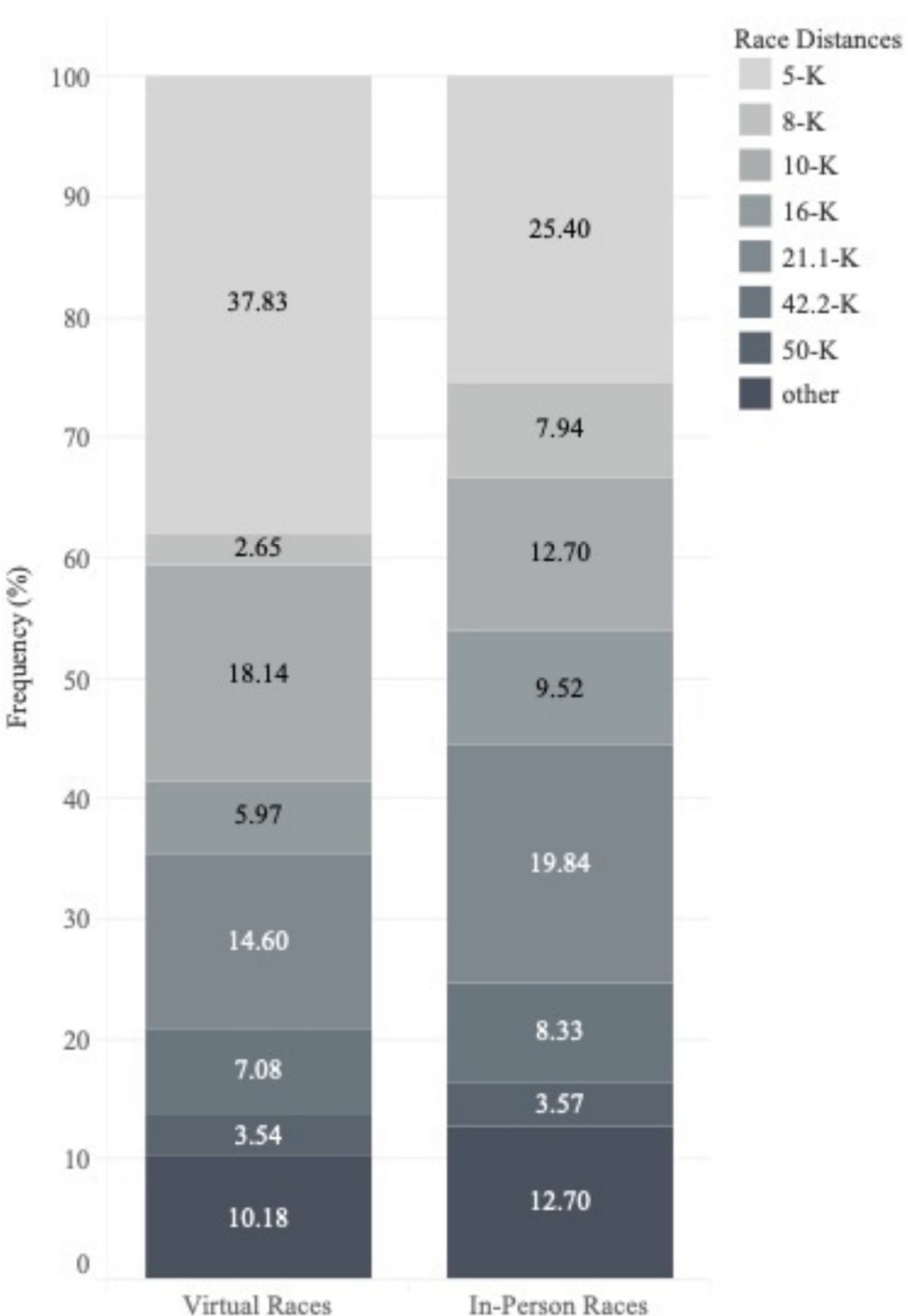

**Fig 3. Virtual and in-person racing trends during the COVID-19 pandemic.** Stacked bar plots depicting virtual (left graph) and in-person (right graph) race participation during the pandemic. Response percentages are listed by response category overlaid on the bar plots.

**Table 2. Logistic regression outcomes comparing running behaviors during 1 year of the pandemic to the year prior to the pandemic.**

| Predictor | Comparison | Total N Runs Odds Ratio (95% CI) | | Sustained Runs Odds Ratio (95% CI) | | Workouts Odds Ratio (95% CI) | | Cross-Training Odds Ratio (95% CI) | | Times of Day Odds Ratio (95% CI) | | Running Motives Odds Ratio (95% CI) | |
|---|---|---|---|---|---|---|---|---|---|---|---|---|---|
| | | ↓ vs. No Change | ↑ vs. No Change | ↓ vs. No Change | ↑ vs. No Change | ↓ vs. No Change | ↑ vs. No Change | ↓ vs. No Change | ↑ vs. No Change | ↓ vs. No Change | ↑ vs. No Change | ↓ vs. No Change | ↑ vs. No Change |
| Age | 18–25 vs. 26–35 | 0.73 (0.34, 1.56) | 1.60 (0.64, 4.04) | 0.84 (0.38, 1.86) | 1.22 (0.50, 3.00) | 0.96 (0.49, 1.87) | 0.46 (0.11, 1.89) | 0.72 (0.34, 1.53) | 1.08 (0.49, 2.37) | *0.30 (0.09, 1.01)** | *2.47 (1.07, 5.67)** | 1.74 (0.90, 3.27) | 0.55 (0.21, 1.43) |
| | 18–25 vs. 36–45 | 1.31 (0.59, 2.91) | *2.79 (1.06, 7.32)** | 1.14 (0.49, 2.64) | 1.93 (0.76, 4.91) | 0.93 (0.45, 1.93) | 1.02 (0.26, 4.00) | *0.47 (0.22, 1.04)** | 0.98 (0.41, 2.35) | *0.23 (0.05, 0.94)** | *4.15 (1.48, 11.63)** | *2.02 (1.07, 4.06)** | 0.38 (0.12, 1.19) |
| | 18–25 vs. 46–55 | *0.35 (0.12, 1.01)** | 1.14 (0.38, 3.39) | 0.80 (0.32, 2.02) | 0.96 (0.33, 2.80) | 0.76 (0.33, 1.73) | 1.26 (0.31, 5.10) | 0.70 (0.30, 1.64) | 0.69 (0.26, 1.84) | 1.35 (0.29, 6.30) | 2.16 (0.82, 5.72) | 1.89 (0.90, 4.01) | 0.27 (0.07, 1.02) |
| | 18–25 vs. 56+ | 0.67 (0.20, 2.31) | 1.02 (0.25, 4.23) | 0.70 (0.20, 2.46) | 1.61 (0.46, 5.62) | 0.93 (0.32, 2.66) | 0.85 (0.08, 9.48) | 0.80 (0.25, 2.51) | 1.54 (0.51, 4.68) | 2.23 (0.43, 11.52) | 1.21 (0.37, 3.99) | *4.69 (1.77, 12.43)** | 0.52 (0.10, 2.67) |
| Sex | Male vs. Female | 1.15 (0.64, 2.06) | 0.88 (0.49, 1.59) | 0.82 (0.47, 1.43) | 1.24 (0.68, 2.27) | 1.62 (0.97, 2.70) | 1.05 (0.40, 2.80) | 0.82 (0.50, 1.35) | 1.51 (0.84, 2.71) | 1.24 (0.53, 2.88) | 1.14 (0.62, 2.11) | 1.08 (0.69, 1.69) | 0.67 (0.33, 1.38) |
| Years Running Experience | 0–3 vs. 4–10 | 2.53 (0.91, 7.01) | 0.55 (0.24, 1.26) | *3.00 (0.99, 9.04)** | 0.72 (0.30, 1.75) | 1.23 (0.60, 2.56) | 0.57 (0.16, 2.11) | 0.56 (0.27, 1.14) | 1.21 (0.48, 3.09) | 2.04 (0.53, 7.89) | 1.57 (0.66, 3.73) | 0.95 (0.49, 1.82) | 2.39 (0.64, 8.94) |
| | 0–3 vs. 11–15 | 2.53 (0.80, 8.02) | 0.50 (0.18, 1.39) | *4.41 (1.32, 14.82)** | *0.42 (0.19, 0.94)** | 1.03 (0.43, 2.51) | 1.34 (0.33, 5.39) | 0.49 (0.20, 1.18) | 1.18 (0.40, 3.52) | 1.43 (0.28, 7.19) | 1.24 (0.43, 3.59) | 1.34 (0.60, 2.96) | 3.42 (0.64, 15.61) |
| | 0–3 vs. 16–20+ | 1.46 (0.46, 4.64) | 0.57 (0.23, 1.40) | 2.32 (0.70, 7.71) | 0.94 (0.36, 2.43) | 0.78 (0.34, 1.80) | 0.24 (0.04, 1.31) | 0.54 (0.24, 1.21) | 1.59 (0.57, 4.45) | 1.88 (0.42, 8.29) | 0.97 (0.35, 2.67) | *2.36 (1.13, 4.96)** | 2.27 (0.51, 10.14) |

Comparison of runners of varying age, sex, and years of running experience on running behaviors during the year of the pandemic to the year prior to the pandemic. Reference levels for the logistic regression were no change from the year prior to the pandemic, runners ages 18–25, males, and runners with 0–3 years of running experience.

Abbreviations: N, number; CI, confidence interval.

*Significant at p < .05.

**Table 3. Running-related injury characteristics during the first full year of the pandemic.**

| | Sprain/ Ligamentous (% by Region) | Strain/ Musculotendinous (% by Region) | Fracture/ Stress Fracture/ Bone Injury (% by Region) | Other (% by Region) | Total Injuries by Location (% of total) |
|---|---|---|---|---|---|
| Toe | 6 (85.7%) | 1 (14.3%) | 0 (0%) | 0 (0%) | 7 (2.6%) |
| Foot | 4 (9.1%) | 16 (36.4%) | 5 (11.4%) | 19 (43.2%) | 44 (16.5%) |
| Ankle | 14 (38.9%) | 16 (44.4%) | 1 (2.8%) | 5 (13.9%) | 36 (13.5%) |
| Lower Leg | 0 (0%) | 28 (63.6%) | 9 (20.5%) | 7 (15.9%) | 44 (16.5%) |
| Knee | 4 (9.3%) | 22 (51.2%) | 0 (0%) | 17 (39.5%) | 43 (16.1%) |
| Thigh | 1 (9.1%) | 5 (45.5%) | 1 (9.1%) | 4 (36.4%) | 11 (4.1%) |
| Hamstring | 0 (0%) | 23 (92%) | 0 (0%) | 2 (8%) | 25 (9.4%) |
| Hip | 3 (8.6%) | 19 (54.3%) | 1 (2.9%) | 12 (34.3%) | 35 (13.1%) |
| Groin | 0 (0%) | 0 (0%) | 1 (33.3%) | 2 (66.7%) | 3 (1.1%) |
| Abdomen | 0 (0%) | 0 (0%) | 1 (50%) | 1 (50%) | 2 (0.7%) |
| Lower Back | 0 (0%) | 10 (58.8%) | 2 (11.8%) | 5 (29.4%) | 17 (6.4%) |

Percentages of running-related injuries during the first full year of the pandemic by body region and injury type.

We identified that runners increased their total running volume during the first full year of the pandemic compared to both the month of eased restrictions, and the year prior to the pandemic. These findings align with other running-related surveys conducted earlier in 2020 that identified significantly altered training behaviors in adult and youth runners in the first several months of the pandemic compared to either 1-month [8], 6-months [9], or the full year prior to social isolation [7]. Our findings support that running habits adapted earlier in the pandemic persisted for the full year of restrictions, despite the fact that about 50% of the polled sample lost access to typical outdoor training environments, and up to 99% lost access to indoor training sites. Respondents also overwhelmingly reported reduced motives to continue training [7]. While cardiovascular exercise is linked with improved mood and mental health [5], excessive physical activity may lead to symptoms of overtraining and depressive symptoms [6]. Higher training volume coupled with changes in external stressors have previously been documented as contributors to increased injury risk [12–14]. These relationships have been substantiated with recent pandemic-related running research findings [7–9], and with the relationships between loss of access to preferred training environments to running-related injuries we identified in this sample.

The benefit of conducting this follow-up survey was that we were able to directly compare the first full years' worth of injury data prior to and during the pandemic instead of adjusting injury risk ratios for exposure time [7]. In addition to loss of access to preferred training environments, novice runners were more likely to sustain a running-related injury during the first full year of the pandemic. There is substantial research supporting that runners with minimal experience are more prone to developing running-related injuries [14–17], which may be attributed to training errors and lack of knowledge about training parameters, such as volume and intensity as they relate to training stress. Previous work assessing inexperienced runners have identified that making more simultaneous changes in training factors, such as mileage, distance, and intensity, was associated with higher injury risk compared to making modest adjustments [18]. While it was not possible to control for local guidelines limiting runners' access to preferred training routes, it is plausible that more experienced runners were more resilient to modest changes in running training. Proper education for inexperienced runners regarding training volume and implications for health are imperative to mitigate running-related injuries, particularly in light of restrictions being lifted and access to a wider variety of training surfaces is becoming available. Clinicians who care for injured runners should also be aware of the higher patient load following the pandemic, and should be prepared to connect with key stakeholders to ensure a modest transition back to sport to reduce future injury rates.

During the month of eased restrictions, respondents decreased running volume and cited fewer opportunities to run throughout the day compared to the pandemic. We postulate that these changes are attributed to lifestyle and demographic factors, as these have been noted as barriers to running participation prior to the pandemic [8]. Broad comparisons across time-points yielded significant yet small effect sizes that may be less clinically meaningful when considering the running population at large, underscoring the need to look at mediating factors that influence running behaviors. Unfortunately, we were unable to definitively determine what influenced our survey responses as we did not re-collect demographic outcomes. However, in the subset of participants that provided identical e-mail addresses to match past responses associated with demographic data, we identified participants' age as a contributing factor to running volume [7]. Several participants offered short-responses at the end of the survey, and cited work, childcare, and life changes (i.e., relocating, injury, general health) as barriers to continuing to run, which would explain the changes in the middle-aged runner groups. Other recent work has identified time as a substantial barrier to running participation [19, 20]. Therefore, although social isolation measures are lifting, other external obligations should be

acknowledged as potential factors influencing reduced running participation. Changing multiple running behaviors, either inadvertently or purposefully, has been identified as one component in the complex running-related injury framework [13, 14, 21]. Runners should be made aware of the influence of running behaviors on potential injury moving forward, and clinicians should be aware of the bearing this may have on future injury epidemiology that may influence patient loads following the pandemic. This is particularly salient information for clinicians should be aware of, considering that we identified a higher proportion of time-loss injuries during the year of the pandemic, suggesting higher injury severity, that often necessitate clinical care.

Based on our descriptive assessments, runners reported higher socialization motives compared to the first several months of the pandemic (9% versus 4%), however socialization did not fully return to pre-pandemic levels (9% versus 11%), with only about half of the sample being able to return to their preferred runner groups. These findings suggest that key social aspects of running for exercise may still be affected despite easing social isolation restrictions. Social isolation measures resultant of the pandemic were found to have profound negative influences on the general populations' mental health, particularly among individuals with low psychosocial support [22]. While moderate cardiovascular exercise was found to mitigate the negative effects of social isolation on mental health during the pandemic [10, 23], the inability to connect with peers through running groups may have lingering implications for mental health and extrinsic motivation to continue running as exercise [19].

Novice runners were more likely to decrease their number of sustained runs per week, and cited fewer motives to continue running as exercise compared to experienced runners, as they may have had less intrinsic reasons for participation and relying more heavily on social support [24]. Virtual social connection opportunities may be helpful for runners with less experience to increase motivations for running training while group running environments remain limited. While we previously identified that the majority of polled runners utilized technology to record their runs [7], we did not assess if runners utilized these technologies for the social capabilities. However, utilizing technology during running has been found to increase external motivations through social running tracking platforms, such as Strava or Map My Run, to connect with peers [25]. Additionally, approximately 50% of the polled sample reported that they virtually participated in races, supporting the positive influence of technology on social and competitive aspects of running during the pandemic.

The majority of virtual races completed during the pandemic were between 5-K and 10-K distances (59%), coinciding with nationwide racing trends in the year prior to the pandemic [2]. However, the subset of respondents who ran in-person races most often competed in longer distances, ranging from 16-K races to ultra-marathons (41%). This shift may be due to the ability to more easily stagger race participants over longer distances compared to shorter courses, however more comprehensive assessments of racing trends and course attributes are needed to assess pandemic-induced changes to competitive running.

There were several limitations to this assessment. There is the potential for recall bias regarding behaviors during the full year of the pandemic given the study design. We were unable to assess re-injuries or chronic injuries explicitly in this study which may have influenced the running-related injury findings, however we accounted for previous injury as a covariate in regression analyses. We did not repeat demographic assessments in this follow-up survey given that we collected these data from the first survey. We received approximately a 40% response rate, and therefore our sample may not reflect the running population at large. Local regulations undoubtedly influenced social isolation restrictions thereby affecting running training, and the extent of eased restrictions experienced by participants. Running trends identified in this survey may not be fully extrapolated to nation- or world-wide behavioral changes.

## Conclusions

During the COVID-19 pandemic, runners were found to increase their total running volume and number of times of day they ran, yet had decreased motives to run compared to both the month of eased restrictions, and the year prior to the pandemic. Younger runners reported more opportunities to run during the day and increased running volume during the pandemic, while inexperienced runners reported fewer motives to run and a higher likelihood to reduce their sustained runs per week. The majority of runners lost access to preferred running locations and running groups during the pandemic, and about half of runners were still unable to return to previous locations and behaviors. Running-related injury risk was overall higher during the first full year of the pandemic, and was substantially higher for runners with minimal experience and runners who lost access to preferred training locations. These findings highlight key changes among the running community influenced by the COVID-19 pandemic that are likely to influence behaviors and RRI as social isolation measures continue to adjust.

## Supporting information

**S1 File. Follow-up COVID-19 running survey.**
(DOCX)

## Author Contributions

**Conceptualization:** Alexandra F. DeJong Lempke, Jay Hertel.

**Data curation:** Alexandra F. DeJong Lempke.

**Formal analysis:** Alexandra F. DeJong Lempke.

**Investigation:** Alexandra F. DeJong Lempke, Jay Hertel.

**Methodology:** Alexandra F. DeJong Lempke, Jay Hertel.

**Project administration:** Alexandra F. DeJong Lempke, Jay Hertel.

**Supervision:** Jay Hertel.

**Visualization:** Alexandra F. DeJong Lempke.

**Writing – original draft:** Alexandra F. DeJong Lempke.

**Writing – review & editing:** Alexandra F. DeJong Lempke, Jay Hertel.

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
