## [Decision Letter · Decision Letter 0]

10 Dec 2021

PONE-D-21-32071Influence of the COVID-19 Pandemic on Running Behaviors, Motives, and Running-Related Injury: A One-Year Follow-Up SurveyPLOS ONE

Dear Dr. DeJong Lempke,

Thank you for submitting your manuscript to PLOS ONE. After careful consideration, we feel that it has merit but does not fully meet PLOS ONE’s publication criteria as it currently stands. Therefore, we invite you to submit a revised version of the manuscript that addresses the points raised during the review process.

Please, moderate your language and be objective with your results.

We look forward to receiving your revised manuscript.

Kind regards,

Daniel Boullosa

Academic Editor

PLOS ONE

Journal Requirements:

Reviewers' comments:

Reviewer's Responses to Questions

**Comments to the Author**

1. Is the manuscript technically sound, and do the data support the conclusions?

Reviewer #1: Yes

Reviewer #2: Partly

2. Has the statistical analysis been performed appropriately and rigorously? 

Reviewer #1: Yes

Reviewer #2: No

3. Have the authors made all data underlying the findings in their manuscript fully available?

Reviewer #1: No

Reviewer #2: Yes

4. Is the manuscript presented in an intelligible fashion and written in standard English?

Reviewer #1: Yes

Reviewer #2: Yes

5. Review Comments to the Author

Reviewer #1: Summary: The purpose of this paper is to determine how full isolation has affected the running community—and better understand changes to motives and behaviors and how these relate to injury. The authors should be commended on their efforts. This is an important study across two timepoints, and their results will have great impact on better understanding running injury risk.

Specific Comments:

Line 80 : Along with other lines in the text, phrasing such as entire year of the pandemic, makes it sound like it was only a year long. I suggest rephrasing so it acknowledges that the pandemic lasted longer than a year. I think technically the current time frame is still in a pandemic.

Line 98: Do you mean that you asked participants to recall the first 3 months? I don’t think May was the initial part of the isolation for the states.

Line 112: Were restrictions lifted everywhere? Do you need this part of the sentence?

Line 113: Please provide definitions of each outcome measure, perhaps in table form.

Line 126: how were bins and categories determined? Why one-unit?

Line 130-131: How were age and years of experience determined?

Line 131: How is 0 a response based on the inclusion criteria?

Line 134: I am bringing this up again, but how do you know month of eased restrictions was the same for everyone? Some locations eased well before others, from my observations.

Line 147: You probably mean present biological sex at time of survey, but I wanted to confirm it was this rather than sex at birth?

Line 162-163: Please provide ranges for these data (age and experience) as well.

Line173-176: While they lost access to gyms and tracks, was this their primary environment for running pre-pandemic?

Table 1. I think for some of these numbers, you could round to the whole number, right? E.g. Total runs per week. Also, your effect sizes should be brought up more in the discussion. Perhaps not directly, but at least referenced.

Reviewer #2: The purpose of the study was to assess long-term implications of pandemic restrictions as well as implications after restrictions were eased.

I commend the authors for investigating long-term implications of pandemic restrictions on runners and feel this is a novel and worthwhile topic. I do have major questions regarding the statistical design, presentation of the results, and interpretation of the results. Please see detailed comments and questions below.

ABSTRACT

General Comments

Results provided in the abstract are different than what are provided in the text of the manuscript.

INTRODUCTION

Specific Comments

L72-73: Consider adding the number of cases during the initial survey date along with the 12.8x higher.

L73: Should this be limited vaccination availability?

L92-93: What specific changes in running behavior do you expect will contribute to higher risk of RRI? Could the increase in running volume be offset by the decrease in training intensity so the risk of RRI is unchanged?

METHODS

General Comment

Did all respondents have eased restrictions in May 2021?

Why were only summary statistics used for assessing running behaviors, motives, and injuries? Consider conducting Fisher’s exact test to compare frequencies of these parameters among the three time points.

Why were two separate paired t-tests conducted? It seems that a repeated measures ANOVA is more appropriate to compare the three time periods.

Specific Comments

L101-104: Were participants contacted twice even if they completed the survey after the initial invite? If so, were there any duplicate responses? Or was the initial contact for the full year of the pandemic and the second contact for the month of eased restrictions?

L106: Consider specifying that this was the original survey.

L113: What is the difference between a sustained run and a workout run?

L127-128: What was the reasoning for using different binning methods (one-unit change vs set criteria)? Could a percent increase/decrease be used instead?

L157: Were only mileage and number of runs included in the binary logistic regression model? What about other running behaviors such as number of workout runs and cross-training activities?

RESULTS

General Comments

Consider reporting effect sizes within the text.

What were the values used for one-unit change for decrease, no change, and increase in running parameters used for the logistic regression?

Figure 1C: Please clarify that this is in response to if they were able to return to their preferred running location(s).

Specific Comments

L171-172: Please describe what differences were found for running time blocks and motives in the text. The bar graphs look like they have very similar distributions.

L177-179: Are you comparing training changes during the year compared to during the month of reduced restrictions? If so, what statistical test was conducted?

L183-184: This seems to fit earlier in the results when you have already started to discuss Figure 1.

L189-191: Please provide specific values and differences.

L195-198: It does not seem appropriate so say injury risk was increased since the difference was non-significant.

DISCUSSION

General Comments

While there were significant differences, most of the effect sizes were small/negligible. Are the differences that you found clinically meaningful?

Specific Comments

L246-248: Is there a known higher patient load? Just because injury risk is higher, is it known if runners are seeking treatment during the pandemic?

L267-269: Are these values statistically significance? Please provide in results section.

L310-311: This is a new findings statement that was not presented in the results or discussion.

6. PLOS authors have the option to publish the peer review history of their article (what does this mean?). If published, this will include your full peer review and any attached files.

Reviewer #1: No

Reviewer #2: **Yes: **Micah Garcia

---

## [Author Response · Author response to Decision Letter 0]

16 Dec 2021

Journal Requirements:

Response: Thank you, we have adjusted the formatting according to these guidelines.

Response: We included full ethics information on lines 80-82.

Reviewers' comments:

Reviewer's Responses to Questions

Comments to the Author

Reviewer #1: Summary: The purpose of this paper is to determine how full isolation has affected the running community—and better understand changes to motives and behaviors and how these relate to injury. The authors should be commended on their efforts. This is an important study across two timepoints, and their results will have great impact on better understanding running injury risk.

Response: We thank you for taking the time to provide a thorough and thoughtful review for this manuscript. We have made every effort to address the concerns specified below.

Specific Comments:

Line 80 : Along with other lines in the text, phrasing such as entire year of the pandemic, makes it sound like it was only a year long. I suggest rephrasing so it acknowledges that the pandemic lasted longer than a year. I think technically the current time frame is still in a pandemic.

Response: We thank you for this comment and agree it is important to acknowledge that changes are on-going. We changed the language to read “the first full year of the pandemic” throughout the entire manuscript, with changes highlighted in red text.

Line 98: Do you mean that you asked participants to recall the first 3 months? I don’t think May was the initial part of the isolation for the states.

Response: We apologize for the confusion; May-June is when we were collecting responses regarding behaviors from March-May. We specified this on line 71.

Line 112: Were restrictions lifted everywhere? Do you need this part of the sentence?

Response: We adjusted our language here, and restrictions were eased across the United States. While we cannot confirm that all respondents were affected similarly, it is important to acknowledge that this was the timeframe in which gyms and tracks, for example, were beginning to re-open. We changed the language to “eased restrictions” on lines 88-89. We included language about this in the limitations as well on lines 369-370.

Line 113: Please provide definitions of each outcome measure, perhaps in table form.

Response: Thank you for this suggestion; however, we provided this information in the supplementary file containing the survey. 

Line 126: how were bins and categories determined? Why one-unit?

Response: We opted to bin these responses as such to represent an increase, decrease, or no change. We used these categories to stay consistent with how we previously reported these outcomes to enable a more direct comparison to past findings.

Line 130-131: How were age and years of experience determined?

Response: These questions were included in the original survey: “demographic data was carried over from the original survey…” on lines 107-108.

Line 131: How is 0 a response based on the inclusion criteria?

Response: In the original survey, participants could have just begun running during the pandemic as a part of our inclusion criteria (“novice runners”).

Line 134: I am bringing this up again, but how do you know month of eased restrictions was the same for everyone? Some locations eased well before others, from my observations.

Response: We appreciate this comment; we addressed this concern in our previous response.

Line 147: You probably mean present biological sex at time of survey, but I wanted to confirm it was this rather than sex at birth?

Response: We phrased this question as biological sex assigned at birth, and included this language on lines 129.

Line 162-163: Please provide ranges for these data (age and experience) as well.

Response: We included these ranges on line 145.

Line173-176: While they lost access to gyms and tracks, was this their primary environment for running pre-pandemic?

Response: Unfortunately, we did not collect that information in the survey; we only specified if they primarily ran indoors, outdoors, or a combination.

Table 1. I think for some of these numbers, you could round to the whole number, right? E.g. Total runs per week. Also, your effect sizes should be brought up more in the discussion. Perhaps not directly, but at least referenced.

Response: Thank you for this suggestion; we would prefer to keep the decimal places so as not to wash out the interpretation of the findings (for example, if we rounded for total number of runs per week, this would not make sense to the reader as to where the differences were noted). We included information about the effect sizes in the discussion on lines 312-315.

Reviewer #2: The purpose of the study was to assess long-term implications of pandemic restrictions as well as implications after restrictions were eased.

I commend the authors for investigating long-term implications of pandemic restrictions on runners and feel this is a novel and worthwhile topic. I do have major questions regarding the statistical design, presentation of the results, and interpretation of the results. Please see detailed comments and questions below.

Response: We thank you for taking the time to provide a thorough and thoughtful review for this manuscript. We have made every effort to address the concerns specified below.

ABSTRACT

General Comments

Results provided in the abstract are different than what are provided in the text of the manuscript.

Response: We apologize, we originally listed differences between one full year of the pandemic compared to the year prior, however we acknowledge that our primary findings compare one full year of the pandemic to the month of eased restrictions. We made the appropriate changes with specific language in the abstract on lines 7-8, 9, 11-12, and 13-14.

INTRODUCTION

Specific Comments

L72-73: Consider adding the number of cases during the initial survey date along with the 12.8x higher.

Response: We included this value on lines 42-43.

L73: Should this be limited vaccination availability?

Response: We added this language on line 44.

L92-93: What specific changes in running behavior do you expect will contribute to higher risk of RRI? Could the increase in running volume be offset by the decrease in training intensity so the risk of RRI is unchanged?

Response: We clarified that we expected that increased training volume would contribute to this; although intensity decreased, we anticipated that overall increased time under tension would relate to increased RRI risk. We also believe that broad changes in running routines and behaviors could contribute to increased injury risk, especially as changes were abrupt and not due to training cycles, etc. We included this more specific hypothesis on lines 64-66.

METHODS

General Comment

Did all respondents have eased restrictions in May 2021?

Response: Thank you for this comment; while we did not assess exact changes in eased restrictions by participant, we believe it is important to highlight that across the US this was the timeframe in which gyms and tracks, for example, were beginning to re-open. We included language about this in the limitations as well on lines 369-370.

Why were only summary statistics used for assessing running behaviors, motives, and injuries? Consider conducting Fisher’s exact test to compare frequencies of these parameters among the three time points.

Why were two separate paired t-tests conducted? It seems that a repeated measures ANOVA is more appropriate to compare the three time periods.

Response: Thank you for this comment; while we intended to perform these analyses, unfortunately we were not able to do so given that not all participants provided matching email addresses to link responses from the initial survey to the follow-up survey. We would only be able to complete the Fisher’s exact tests and RMANOVAs for a subset of participants, which would limit sample size. As such, we kept these assessments separate to be representative of respondents from the current survey. We clarified that this approach was taken due to this issue on lines 113-114, and line 122.

Specific Comments

L101-104: Were participants contacted twice even if they completed the survey after the initial invite? If so, were there any duplicate responses? Or was the initial contact for the full year of the pandemic and the second contact for the month of eased restrictions?

Response: All respondents were contacted twice to complete the survey, however were told to disregard the follow-up email if they completed the original survey link. There were no duplicate responses. We clarified this point on lines 78-79.

L106: Consider specifying that this was the original survey.

Response: We included this language on line 80.

L113: What is the difference between a sustained run and a workout run?

Response: We specified with examples on lines 90-91.

L127-128: What was the reasoning for using different binning methods (one-unit change vs set criteria)? Could a percent increase/decrease be used instead?

Response: We opted to bin these responses to represent an increase, decrease, or no change. While we agree that percentage increase/decrease could be used, we would prefer to stay consistent with how we previously reported these outcomes to enable a more direct comparison to past findings.

L157: Were only mileage and number of runs included in the binary logistic regression model? What about other running behaviors such as number of workout runs and cross-training activities?

Response: The complete list of variables included in the logistic regression model can be found in Table 2, and include total runs per week, sustained runs, workouts, cross-training, training times of day, and running motives. We additionally included this information in the text now on lines 127-128.

RESULTS

General Comments

Consider reporting effect sizes within the text.

Response: We included effect sizes in the results now on lines 152-153.

What were the values used for one-unit change for decrease, no change, and increase in running parameters used for the logistic regression?

Response: These responses were dummy-coded as -1 (decrease), 0 (no change), and 1 (increase) in the logistic regression, listed now on line 104.

Figure 1C: Please clarify that this is in response to if they were able to return to their preferred running location(s).

Response: We included that this was their preferred running locations in the caption now for Figure 1C (lines 168 & 172).

Specific Comments

L171-172: Please describe what differences were found for running time blocks and motives in the text. The bar graphs look like they have very similar distributions.

Response: We apologize for this confusion in wording on our part; the total number of times that individuals would opt to run, and the number of reported motives were different between timepoints, however the distribution of preferred timing and types of motives were similar. We adjusted this sentence on lines 154-156.

L177-179: Are you comparing training changes during the year compared to during the month of reduced restrictions? If so, what statistical test was conducted?

Response: We simply asked participants to indicate if they were able to return to their preferred training locations, and as such this is a percentage of the respondent pool, not a statistical test. We adjusted the language for clarity on line 181.

L183-184: This seems to fit earlier in the results when you have already started to discuss Figure 1.

Response: We moved this sentence up now so that it directly following citing Figure 1c, and can be found now on lines 179-181.

L189-191: Please provide specific values and differences.

Response: We included these details now on lines 220-243.

L195-198: It does not seem appropriate so say injury risk was increased since the difference was non-significant.

Response: We changed the language accordingly on lines 248-249.

DISCUSSION

General Comments

While there were significant differences, most of the effect sizes were small/negligible. Are the differences that you found clinically meaningful?

Response: We agree that the small effect sizes when looking at broad changes is less clinically-meaningful than considering the effects of demographic factors on training parameters as these undoubtedly influenced changes in behaviors. We incorporated this into the discussion now on lines 312-315.

Specific Comments

L246-248: Is there a known higher patient load? Just because injury risk is higher, is it known if runners are seeking treatment during the pandemic?

Response: While we do not have the exact data on patient loads, we justified this statement based on our findings of more time-loss injuries indicative of potentially more severe injury cases that often necessitate clinical care, found on lines 329-331.

L267-269: Are these values statistically significance? Please provide in results section.

Response: These findings were based upon our descriptive analyses, clarified now on line 332.

L310-311: This is a new findings statement that was not presented in the results or discussion.

Response: These outcomes were previously solely presented in Table 2, however based on your previous suggestions, are now included in the results on lines 220-243. We discussed the influence of demographics in paragraph 4 of the discussion (lines 309-331), and running experience in paragraph 6 (lines 343- 354).

---

## [Decision Letter · Decision Letter 1]

9 Feb 2022

Influence of the COVID-19 pandemic on running behaviors, motives, and running-related injury: a one-year follow-up survey

PONE-D-21-32071R1

Dear Dr. DeJong Lempke,

We’re pleased to inform you that your manuscript has been judged scientifically suitable for publication and will be formally accepted for publication once it meets all outstanding technical requirements.

Kind regards,

Daniel Boullosa

Academic Editor

PLOS ONE

Additional Editor Comments (optional):

Reviewers' comments:

Reviewer's Responses to Questions

**Comments to the Author**

1. If the authors have adequately addressed your comments raised in a previous round of review and you feel that this manuscript is now acceptable for publication, you may indicate that here to bypass the “Comments to the Author” section, enter your conflict of interest statement in the “Confidential to Editor” section, and submit your "Accept" recommendation.

Reviewer #1: All comments have been addressed

Reviewer #2: All comments have been addressed

2. Is the manuscript technically sound, and do the data support the conclusions?

Reviewer #1: Yes

Reviewer #2: Yes

3. Has the statistical analysis been performed appropriately and rigorously? 

Reviewer #1: Yes

Reviewer #2: Yes

4. Have the authors made all data underlying the findings in their manuscript fully available?

Reviewer #1: No

Reviewer #2: No

5. Is the manuscript presented in an intelligible fashion and written in standard English?

Reviewer #1: Yes

Reviewer #2: Yes

6. Review Comments to the Author

Reviewer #1: (No Response)

Reviewer #2: Thank you for addressing the previous comments. I believe the revisions improve the clarity of the manuscript. Below are a couple of minor suggestions to consider.

INTRODUCTION

L44-45: Please report dates in the same format and add a year for the initial collection.

METHODS

L114-116: The location of this text confused me as it talked the following sentence was specific to the follow-up survey and then sentence said they couldn’t be matched. This can likely be removed and merged with L123-124.

L156-157: Consider being specific to the direction of the difference (e.g., more running motives during the month of eased restriction).

7. PLOS authors have the option to publish the peer review history of their article (what does this mean?). If published, this will include your full peer review and any attached files.

Reviewer #1: No

Reviewer #2: **Yes: **Micah Garcia

---

## [Editor Report · Acceptance letter]

7 Mar 2022

PONE-D-21-32071R1 

Influence of the COVID-19 pandemic on running behaviors, motives, and running-related injury: a one-year follow-up survey 

Dear Dr. DeJong Lempke:

I'm pleased to inform you that your manuscript has been deemed suitable for publication in PLOS ONE. Congratulations! Your manuscript is now with our production department. 

Kind regards, 

on behalf of

Dr. Daniel Boullosa 

Academic Editor

PLOS ONE